# Acceptability of pre-exposure prophylaxis and associated factors among HIV-negative young men in Kagwara fishing community-Serere district, Uganda: A cross-sectional study

Alex Omoding[1,2], Ronald Opito [3]*, Paul Oboth[4], Francis Okello[1], Joseph K. B. Matovu[1,5]

**1** Department of Community and Public Health, Faculty of Health Sciences, Busitema University, Mbale, Uganda, **2** Department of Health, Uganda Protestant Medical Bureau (UPMB), Kampala, Uganda, **3** Department of Public Health, School of Health Sciences, Soroti University, Soroti, Uganda, **4** Department of Biochemistry and Molecular Biology, Faculty of Health Sciences, Busitema University, Mbale, Uganda, **5** Department of Disease Control and Environmental Health, Makerere University School of Public Health, Kampala, Uganda

* dopito@sun.ac.ug

## Abstract

### Background

Despite the potential efficacy of Pre-Exposure Prophylaxis (PrEP) in reducing HIV risk, Oral PrEP acceptability remains strikingly varied by populations and locations. We assessed PrEP acceptability and associated factors among at-risk HIV-negative young men.

### Methods

A cross-sectional analytical study design was used. Data were collected among 409 at-risk HIV-negative young men aged 15–24 years living in Kagwara fishing community-Serere district, Uganda between August and October 2023. Quantitative data were collected on socio-demographic characteristics, sexual risk behaviors and concerns about PrEP. Data was analyzed using Stata version 15.0 statistical software. Summary statistics were computed and presented as tables, frequencies and proportions. Bivariate analysis was conducted using binary logistic regression to identify independent factors associated with PrEP acceptability. All factors that had $p < 0.10$ at the bivariate analysis and confounders were entered into the final logistic regression model. All factors with $p < 0.05$ were considered significantly associated with the primary outcome.

### Results

The average age of 409 respondents was 21.8 (Standard Deviation [SD]=1.9) years. Majority, (97.8%, n = 393) had unprotected penetrative sex. PrEP acceptability was

**Data availability statement:** All relevant data are within the manuscript and its Supporting Information files.

**Funding:** The author(s) received no specific funding for this work.

**Competing interests:** The authors have declared that no competing interests exist.

**Abbreviations:** AGYW, adolescent girls and young women; AIDS, Acquired Immunodeficiency Syndrome; AVERT, Aid Virus Education and Research Trust; AYFRHS, Adolescent and Youth Friendly Reproductive Health Care Services; CDC, Centers for Disease Control and Prevention; DHIS2, District Health Information System Software version 2; HIV, Human Immunodeficiency Virus; HMIS, Health Management Information System; MSM, Men who have sex with Men; MSMW, Men who have Sex with Men and Women; NRTIs, Nucleotide Reverse Transcriptase Inhibitors; PrEP, pre-exposure prophylaxis; PWID, People Who Inject Drugs; PWUD, People Who Use Drugs; STD, Sexually Transmitted Disease (s); UN, United Nations; UNAIDS, United Nations Joint Program on HIV; UNFPA, United Nations Fund for Population Activities; UNICEF, United Nations International Children's Emergency Fund; WHO, World Health Organization.

high as majority of the participants accepted to use PrEP based on the six constructs of acceptability (93.6%, n = 383). Participants with perceived risk of getting HIV infection had higher odds of PrEP acceptability, (adjusted odds ratio [aOR]=4.23, 95%CI = 1.05–17.04). Participants who knew their partner's HIV status (aOR=0.25, 95%CI = 0.07–0.88), those who felt embarrassed to ask for PrEP from the facility (aOR=0.12, 95%CI = 0.04–0.39), and those who had stigma associated with use of PrEP (aOR=0.13, 95%CI = 0.04–0.41) had reduced odds of PrEP acceptability.

## Conclusion

We found a high level of PrEP acceptability among young men at risk of HIV acquisition in Kagwara fishing community. Improving access to PrEP services among high risk young men in the fishing communities may increase PrEP uptake in this population and across similar settings. The Ministry of health needs to use multiple approaches to provide PrEP such as peer-led models, drug distribution points, short message reminders for refills, pharmacies and retail drug shops.

## Introduction

Despite the availability of recognized and recommended oral Pre-Exposure Prophylaxis (PrEP) for HIV prevention among at-risk populations, new HIV infections have persisted and are highest among adolescents and young people [1]. Globally, 1.3 million people acquired new HIV in 2023 [2]. In Uganda, an average of 52,000 people become infected with HIV every year, with 36% of these new infections occurring among adolescents and young people [3]. The HIV prevalence in Uganda has a geographical heterogeneity with fishing communities listed among those with higher prevalence as high as 20% or more, against the national prevalence of 5.8% [4–6]. The HIV prevalence among young fisher-folks is 3–4 times higher than the 1.8% among young people in the general population [6,7]. People in the fishing communities have increased susceptibility to HIV due to complex interacting factors like high mobility, poor access to healthcare including HIV prevention and care services, low perception of HIV risk, and poor access to HIV information [8,9]. Also, fishing communities are known for having persons who engage in risky sexual intercourse and transactional sex [10–12], to which some young people get inclined [13,14].

Uptake of oral PrEP among at-risk people in the fishing communities remains largely undocumented, yet its success as an intervention greatly depends on its acceptability [15,16]. According to studies across several populations, PrEP acceptability has varied from as low as 1% among the young people in South Africa [17] to as high as above 90% among the female sex workers in Zambia [18]. Most PrEP implementation efforts have focused primarily on reaching at HIV high-risk populations, especially female sex workers, men who have sex with men, HIV discordant couples, adolescent girls and young women because they account for the majority of new HIV infections [18–22]. Subsequent studies on PrEP use in settings providing standard of care have also focused on the same populations leaving out young men

who are also sexually active and are sexual counterparts of women who are at high-risk of HIV infection. Therefore, this study aimed at assessing acceptability of pre-exposure prophylaxis and associated factors among at-risk HIV-negative young men in Kagwara fishing community- Serere district, Uganda to provide supportive evidence towards provision and scale up of PrEP use.

## Materials and methods

### Study design

This was a cross-sectional analytical design, which involved a quantitative method of data collection and analysis. The design was chosen because it incorporated the use of questionnaire as the data capture tool, with the implication that it was possible to collect quantifiable data that was required to answer the research questions. In addition, collection of not only descriptive data that could be analyzed but also the possibility of analyzing relationships between any descriptive variables of interest [23,24]. In other words, the design allowed for the analysis of all possible factors associated with PrEP acceptability among young men at risk of HIV acquisition.

### Study setting

This study was conducted in Kagwara fishing community, Serere district-Uganda. This fishing community was chosen as the study site because just like other fishing communities, it has high HIV incidence [25]. The ministry of health of Uganda is targeting the fishing communities for provision of PrEP since the fisher folks are considered as priority population and at high risk of HIV acquisition [26]. Kagwara is one of the fishing communities in Lake Kyoga found in Eastern part of Uganda. It is being served by Kagwara health centre III (HC III), one of the health facilities in Kadungulu sub-county, Serere district-Uganda.

### Study population (inclusion and exclusion criteria)

The study participants included were young men aged 15–24 years who were sexually active, having one or multiple sexual partners, reporting inconsistent/no condom use with women of unknown HIV status and young men with known HIV- negative test result. Young men who were medically compromised to provide informed consent and those who did not identify with their birth gender were excluded from the study.

### Measurement of variables

**Dependent variable.** The dependent variable (primary outcome) for this study was acceptability of HIV PrEP which was defined and measured as proportion of high-risk young men that would express intention to use PrEP for HIV prevention purpose if made available to them. PrEP acceptability was measured as a composite variable from six constructs of theoretical framework of acceptability [TFA] [27]. These included: affective attitude, burden, perceived effectiveness, intervention coherence, self-efficacy and opportunity cost. Although the acceptability framework includes seven theoretical constructs, we adapted and used six constructs which were more applicable to our study. Each theoretical construct had one question with five responses on a liked scale. The five responses were collapsed into two (2), that is, positive, scored one (1) and negative, scored as zero (0). This helped with the final computations, and it avoided scores cancelling the result. The sum of the six responses per participant was computed and ranged from zero (0) to six (6). Participants with scores of 4–6 were considered as having accepted PrEP while those with scores of 0–3 were considered as having not accepted PrEP.

**Independent variables.** The independent variables in this study were factors associated with PrEP acceptability that included the sociodemographic characteristics such as age (measured as age in completed years), marital status (single, married), education level (non, primary, secondary, tertiary), religion, behavioral characteristics such as having multiple

sexual partners, knowing the partners' HIV status, use of drugs of abuse and other medical related conditions such as having ever been diagnosed with sexually transmitted infection (STI).

## Data collection methods

The study participants were recruited and data collected between August-October 2023. The young men were accessed through the leaders of the fishing community. Their age eligibility (15–24 years) to participate in the study was verified using the documents obtained at the landing site. The 4 research assistants (RAs) who collected the data were graduates in either social sciences or medical sciences, who had experience in field data collection or volunteering in HIV prevention and treatment clinics. They were trained in data collection using online tools, consenting processes and ethical issues during data collection. In addition, they undertook an online course on good clinical practices and obtained certification.

## Participant recruitment

On daily basis working with community leaders, young men found doing activities of the day such as boat cleaning and preparation, fish smoking/drying, selling fish or those on their activities of leisure such as playing cards, watching football or betting at the landing site would be approached and the purpose of the visit explained. Those who accepted to participate in the study would randomly be selected and the subsequent procedures would then be conducted that included further screening for study eligibility, parental consenting for the minors and obtaining informed consent from the participants, offering HIV counselling and testing and those who turned negative would then be interviewed following research questionnaire that had been configured into the online data collection tool, KOBO toolbox (https://www.kobotoolbox.org/). The two participants who were found to be living with HIV were linked to their nearest health facilities (Kagwara Health Centre III) for HIV care.

## Data analysis

Data were extracted from the KOBO toolbox and exported to STATA (version 15.0) statistical software for analysis. Data were analyzed at 3 different levels. Descriptive statistics such as mean, frequencies, standard deviations and range were done and presented as tables, frequencies and proportions. Bivariate analysis was conducted using penalized logistic regression to compare independent variables with the outcome of interest (PrEP acceptability). The final multivariable analysis involved using penalized logistic regression to determine the factors associated with acceptability of PrEP among this population. The independent factors included in the final multivariable model were sociodemographic characteristics (age group, marital status, education level, occupation) and behavioral characteristics (number of sexual partners, perceived risks of HIV infection, use of addictive drugs among others). They were included in the final model based on their plausibility and/or the P-value cut off of 0.10 at bivariate level. These were reported as adjusted odds ratios (aOR) and factors whose confidence intervals of the OR did not include a null (1.0) were considered statistically significant. The level of significance was set at 5%.

## Ethical approval and consent to participate

Ethical approval to conduct the study was obtained from Busitema University Research and Ethics Committee (REC), Approval number **BUFHS-2023-61 on 06/04/2023**. Administrative clearance was obtained from the district health office, Serere district and the local government representative office in whose jurisdiction Kagwara landing site was situated. Each participant at his own discretion provided informed consent (written) as was indicated on the consent form, while parental consent was obtained for all the minors prior to their participation in the study. Confidentiality was maintained by ensuring that individual patient-level data obtained was de-identified, encrypted, and passworded to ensure access by only authorized team of investigators. All interviews were conducted in secure places to ensure the privacy of the study participants during the process.

### Inclusivity in global research

Additional information regarding the ethical, cultural, and scientific considerations specific to inclusivity in global research is included in the Supporting information (S1 Checklist).

## Results

### Socio-demographic characteristics of the respondents

Table 1 shows the socio-demographic characteristics of the study population. Overall, 409 respondents participated in the study. The average age of the respondents was 21.8 (Standard Deviation = 1.9 years. Majority (87.8%, n = 359) of the respondents were aged 20–24 years while most (6 1.9%, n = 253) participants had attained primary education and were fishermen (79%, n = 323).

### Sexual activities and HIV risk prevention awareness of the participants

Table 2 shows sexual activities and HIV risk prevention awareness. Most (59.2% n = 242) of the respondents had multiple sexual partners, majority (97.8%, n = 393) had un-protected sexual intercourse within six months of the study and 84.6% (n = 346) did not know their partner (s) HIV status. Majority (91.9%, n = 376) of the respondents would not feel embarrassed to ask for PrEP from health facilities if made available freely, while most (63.1%,n = 258) respondents would take PrEP because of having multiple sexual partners.

### Concerns about HIV PrEP

Table 3 shows concerns about HIV PrEP highlighted by the young men. Most (72.9%, n = 298) of the participants had concerns about PrEP. The commonly reported concerns were effectiveness issues (55.7%, n = 166), side effects (58.1%, n = 173) and pill burden (39.9%, n = 119).

The numbers in this table vary because the first question was asked to all respondents, and the subsequent questions were asked only respondents who had concerns regarding PrEP.

**Table 1. Socio-demographic characteristics of the respondents.**

| Variables | Frequency | Proportion (%) |
|---|---|---|
| **Mean age/years** | 21.8 (SD ± 1.9) | |
| **Age/ years** | | |
| 15–19 | 50 | 12.2% |
| 20–24 | 359 | 87.8% |
| **Education level** | | |
| None | 64 | 15.6% |
| Primary | 253 | 61.9% |
| Secondary | 88 | 21.5% |
| Tertiary | 4 | 1.0% |
| **Occupation** | | |
| Fisherman | 323 | 79.0% |
| Non-fisherman | 86 | 21.0% |
| **Marital status** | | |
| Single/never married | 129 | 31.6% |
| Married/cohabiting | 280 | 68.4% |

Table 2. Sexual activities and HIV risk prevention awareness of participants.

| Variable | Frequency | Percentage |
|---|---|---|
| **Number of sexual partners** | | |
| One | 167 | 40.8% |
| More than one | 242 | 59.2% |
| **Was penetrative sex protected (N=404)** | | |
| No | 393 | 97.8% |
| Yes | 13 | 3.2% |
| **Know partner HIV status** | | |
| No | 346 | 84.6% |
| Yes | 63 | 15.4% |
| **Used illicit drugs in the last 6 months** | | |
| | 215 | 52.6% |
| Yes | 194 | 47.4% |
| **Ever heard about PrEP** | | |
| No | 250 | 61.1% |
| Yes | 159 | 38.9% |
| **HIV services offered with other services at the facility** | | |
| No | 0 | 0.0% |
| Yes | 409 | 100.0% |
| **Feeling embarrassed to ask for PrEP at facility** | | |
| No | 376 | 91.9% |
| Yes | 33 | 8.1% |
| **Perceived risk of HIV infection motivates me to use PrEP** | | |
| No | 88 | 21.5% |
| Yes | 321 | 78.5% |
| **Having multiple sexual partners motivates me to use PrEP** | | |
| Yes | 258 | 63.1% |
| No | 151 | 36.9% |
| **Having sex without condoms motivates me to use PrEP** | | |
| Yes | 313 | 76.5% |
| No | 96 | 23.5% |
| **Having sexual partners of unknow HIV status motivates me to use PrEP** | | |
| Yes | 321 | 78.5% |
| No | 88 | 21.5% |
| **History of sexually transmitted disease** | | |
| Yes | 391 | 95.6% |
| No | 18 | 4.4% |

## PrEP acceptability based on the constructs of theoretical framework of acceptability (TFA)

Table 4 shows PrEP Acceptability based on 6 constructs of the theoretical framework of acceptability (TFA) as described in the methods sections. Overall, 93.6% (n=383) of the at-risk young men were ready to accept PrEP for HIV prevention purpose. The majority (91.4%, n=373) of the participants felt comfortable using PrEP, while most (76.5%, n=313) felt there was no burden associated with PrEP and strongly mentioned that it will take little or no effort to use PrEP. Majority (92.9%, n=380) of the respondents agreed that it was clear to them how using PrEP reduces chances of getting infected with HIV (intervention coherence).

**Table 3. Concerns about HIV PrEP.**

| Variable | Frequency | Percentage |
|---|---|---|
| **Having any PrEP concerns** | | |
| No | 111 | 27.1% |
| Yes | 298 | 72.9% |
| **Effectiveness of less than 100% (N = 298)** | | |
| No | 132 | 44.3% |
| Yes | 166 | 55.7% |
| **Side effects concern (N = 298)** | | |
| No | 125 | 41.9% |
| Yes | 173 | 58.1% |
| **Cost concerns (N = 298)** | | |
| No | 210 | 70.5% |
| Yes | 88 | 29.5% |
| **Pill burden concern (N = 298)** | | |
| No | 179 | 60.1% |
| Yes | 119 | 39.9% |
| **Stigma concern (N = 298)** | | |
| No | 203 | 68.1% |
| Yes | 95 | 31.9% |
| **Still must use condom concern (N = 298)** | | |
| No | 261 | 87.6% |
| Yes | 37 | 12.4% |
| **Health workers negative receptivity concern (N = 298)** | | |
| No | 294 | 98.7% |
| Yes | 4 | 1.3% |

Note: The numbers in this table vary because the first question was asked to all respondents, and the subsequent questions were asked only respondents who had concerns regarding PrEP.

### Factors associated with PrEP acceptability

Table 5 shows the factors associated with PrEP acceptability among young men in Kagwara fishing community. At a multivariable level, young men who perceived HIV risk had four times higher odds of accepting PrEP compared to their counterparts who never perceived HIV risk, (aOR =4.23, 95%CI = 1.05–17.04). Young men who knew their partners' HIV status had lower odds of accepting PrEP compared to those who did not know their partners HIV status, (aOR=0.25, 95%CI = 0.25 0.07–0.88). Those who felt embarrassed to ask for PrEP from the facility had lower odds of accepting PrEP, (aOR=0.12, 95%CI = 0.04–0.39), and young men who were concerned about stigma of using PrEP had lower odds of accepting PrEP, (aOR=0.13, 95%CI = 0.04–0.41).

### Discussion

This study assessed HIV PrEP acceptability and associated factors among young men at substantial risk of HIV infection in a fishing community. The PrEP acceptability in this community was very high, as majority of participants (93.6%) were willing to take up PrEP for HIV prevention if made available. Previous studies documented low PrEP acceptability among various populations. Among young men, PrEP acceptability ranged from 45.45% to 76.80% in Sub-Saharan Africa (SSA) [28].

In Uganda, PrEP acceptability has been equally low in the earlier studies. For instance the study done in the fishing community of Lake Victoria found PrEP acceptability ranging from 14%−24% [29]. Subsequent studies have shown

**Table 4. PrEP acceptability based on the theoretical framework of acceptability.**

| Construct | Question asked | Response (score) | Frequency | Proportions (%) |
|---|---|---|---|---|
| Affective Attitude | How comfortable do you feel using PrEP? | Comfortable (1) | 374 | 91.4 |
| | | Uncomfortable (0) | 35 | 8.6 |
| Burden | How much effort will it take to use PrEP? | Little/No effort (1) | 313 | 76.5 |
| | | A lot of effort (0) | 96 | 23.5 |
| Perceived Effectiveness | PrEP is likely to reduce my chance of getting infected with HIV | Agree (1) | 337 | 82.4 |
| | | Disagree (0) | 72 | 17.6 |
| Intervention coherence | It is clear to me how using PrEP reduces my chances of getting infected with HIV | Agree (1) | 380 | 92.9 |
| | | Disagree (0) | 29 | 7.1 |
| Self-Efficacy | How confident do you feel that you can do what is required to use PrEP | Confident (1) | 366 | 89.5 |
| | | Unconfident (0) | 43 | 10.5 |
| Opportunity Cost | Using PrEP will interfere with my other priorities | Agree (0) | 37 | 9.1 |
| | | Disagree (1) | 372 | 90.9 |
| PrEP Acceptability | Total of the 6 scores is 4 or more | Yes (Scores≥4) | 383 | 93.6 |
| | | No (Scores<4) | 26 | 6.4 |

increasing PrEP acceptability among various populations. For example Bashir Ssuna et al reported 8 in 10 fisher folks in Kampala were willing to use PrEP [30]. Likewise, Susan S Witte reported 9 in 10 female sex workers were willing to initiate PrEP in Uganda [31]. This higher acceptance could have resulted from sustained behavior change messages targeting high risk communities such as fisher folks in the fishing communities and the adoption and promotion of PrEP as a key biomedical strategy for HIV prevention by ministry of health of Uganda [32]. The acceptability of PrEP by these groups therefore provides an opportunity for the ministry of health to expand access to PrEP services to reach the communities as one of a combination prevention strategy to reaching HIV epidemic control in Uganda.

We found perceived risk of HIV infection as a positive factor associated with increased PrEP acceptability among the at-risk young men. This implies that most of the young men engage in sexual relationships without knowing their partners' HIV status. The perception here can be attributed to limited HIV testing in most of the communities in Uganda and young men practice unsafe sex which results in increased spread of HIV. This finding agrees with other studies that found a relationship between perceived risk of HIV and PrEP acceptability [33–35]. It is important to make PrEP services available along with HIV-tests and build capacity of the young men in performing HIV self-tests. In addition, various HIV testing modalities that are user friendly should be readily available like HIV self-testing at both facility and community.

Our study found knowing partners' HIV status is one of the factors associated with reduced PrEP acceptability. This could be attributed to mutual trust developed by the partners. This finding is contrary with those of Ssuna et al. who did their study in Uganda and found knowledge of HIV status being associated with high PrEP use [30]. They reported that study participants who tested for HIV within the past 6 months were more likely to use PrEP. This requires packaging of HIV prevention messages in focused manner, educating young men about thoughts around HIV window period and use of multiple HIV prevention methods like circumcision to cater for those that will not accept PrEP.

Feeling embarrassed to ask for PrEP from the health facilities was another factor associated with no acceptability of PrEP among the young men who participated in this study. This can be attributed to inferiority and high illiteracy levels common in the fishing communities in Uganda. This finding is in line with the study conducted among transgender women in U.S.A that identified being embarrassed with health care system as one of the factors affecting PrEP acceptability [36]. There is a need to train and orient health workers on the provision of youth friendly health services at both health facilities and communities with focus in places where young men always have leisure activities.

**Table 5. Factors associated with PrEP acceptability among at-risk, HIV-negative men in Kagwara fishing community.**

| Characteristic | Willing to accept PrEP | | Odds Ratio (OR) | | P-value |
| --- | --- | --- | --- | --- | --- |
| | Yes (n, %) 383 (93.6%) | No (n, %) 26 (6.4%) | Crude OR (95%CI) | | Adjusted OR (95%CI) |
| **Age group** | | | | | |
| 15–19 years | 39 (78.0) | 11 (22.0) | 1.00 | 1.00 | |
| 20–24 years | 344 (95.8) | 15 (4.2) | **6.47 (2.82-14.85)** | 2.98 (0.80-11.13) | 0.104 |
| **Highest level of education level attained** | | | | | |
| Primary | 295 (93.1) | 22 (6.9) | 1.00 | 1.00 | |
| Post-primary | 88 (95.7) | 4 (4.3) | 1.49 (0.53-4.24) | 1.00 (0.21-4.85) | 0.995 |
| **Occupation** | | | | | |
| Non-Fisherman | 76 (88.4) | 10 (11.6) | 1.00 | 1.00 | |
| Fisherman | 307 (95.1) | 16 (4.9) | 2.16 (0.90-5.17) | 0.51 (0.12-2.11) | 0.351 |
| **Marital status** | | | | | |
| Single | 111 (86.1) | 18 (13.9) | 1.00 | 1.00 | |
| Married | 272 (97.1) | 8 (2.9) | **5.32 (2.29-12.34)** | 2.64 (0.82-8.49) | 0.103 |
| **Number of sexual partners** | | | | | |
| One | 149 (89.2) | 18 (10.8) | 1.00 | 1.00 | |
| >One | 234 (96.7) | 8 (3.3) | **3.41 (1.48-7.89)** | 3.01 (0.95-9.56) | 0.061 |
| **Known partner HIV status** | | | | | |
| No | 326 (94.2) | 20 (5.8) | 1.00 | 1.00 | |
| Yes | 57 (90.5) | 6 (9.5) | 0.56 (0.22-1.40) | **0.25 (0.07-0.88)** | **0.031** |
| **Used illicit drugs in the last 6 months** | | | | | |
| No | 198 (92.1) | 17 (7.9) | 1.00 | 1.00 | |
| Yes | 185 (95.4) | 9 (4.6) | 1.72 (0.76-3.89) | 1.30 (0.38-4.44) | 0.672 |
| **Ever heard about PrEP** | | | | | |
| No | 229 (91.6) | 21 (8.4) | 1.00 | 1.00 | |
| Yes | 154 (96.9) | 5 (3.1) | **2.63 (1.01-6.86)** | 1.03 (0.29-3.61) | 0.968 |
| **I am feeling embarrassed to ask for PrEP at the facility.** | | | | | |
| No | 364 (96.8) | 12 (3.2) | 1.00 | 1.00 | |
| Yes | 19 (57.6) | 14 (42.4) | **0.05 (0.02-0.11)** | **0.12 (0.04-0.39)** | **0.000** |
| **The perceived risk of HIV acquisition motivates me to use PrEP.** | | | | | |
| No | 81 (92.0) | 7 (8.0) | 1.00 | 1.00 | |
| Yes | 302 (94.1) | 19 (5.9) | 1.43 (0.59-3.43) | **4.23 (1.05-17.04)** | **0.042** |
| **Effectiveness of less than 100%** | | | | | |
| No | 121 (91.7) | 11 (8.3) | 1.00 | 1.00 | |
| Yes | 153 (92.2) | 13 (7.8) | 1.08 (0.47-2.45) | 0.35 (0.10-1.17) | 0.088 |
| **Side effects concern** | | | | | |
| No | 115 (92.0) | 10 (8.0) | 1.00 | 1.00 | |
| Yes | 159 (91.9) | 14 (8.1) | 1.00 (0.44-2.29) | 0.52 (0.18-1.51) | 0.227 |
| **Stigma concern** | | | | | |
| No | 199 (98.0) | 4 (2.0) | 1.00 | 1.00 | |
| Yes | 75 (79.0) | 20 (21.0) | **0.08 (0.03-0.24)** | **0.13 (0.04-0.41)** | **0.001** |

Notes: Bold=Significant with P<0.05, P-values indicated are for aOR.

In addition, our study found concern of stigma associated with PrEP use as another factor that reduced PrEP acceptability among the young men. Stigma has been associated with the reduction of health care acceptability particularly in chronic illness like HIV. This finding has been also reported by study conducted among transgender women in U.S.A [37]. there is a need to break this barrier through continued sensitization of the young men to embrace the available HIV prevention methods.

## Study limitations and strengths

Our study had some limitations as highlighted below. We did not provide physical PrEP to study participants but only reported their presumed acceptability. The observed high acceptability is therefore prone to social desirability bias. The recruitment of only men limited external validity and extrapolation of findings to the general population. Some participants might have had prior knowledge or education on PrEP, and this might have insinuated some responses.

Despite these limitations, our study had strengths: This was among the few studies conducted among young men in the fishing community which is one of the unique populations prone to HIV. The knowledge gained from the present study may aid in lowering HIV infections among the young men, findings can help to inform efforts to increase health promotion and continued sanitization to encourage young men to accept PrEP.

## Conclusion

Our study revealed that HIV PrEP is acceptable among young men at risk of HIV infection. These findings indicate the need to promote HIV PrEP among young men in the fishing communities with intensified health promotion messages. There is a need to continue promoting use of multiple HIV prevention approaches and create awareness of the eligibility for PrEP.

The Ministry of health needs to use multiple approaches to provide PrEP such as peer-led models, drug distribution points, short message reminders for refills, pharmacies and retail drug shops

It is important to identify barriers to PrEP acceptability among this population using a qualitative approach. This will help in designing person centered approach to PrEP delivery among young men in the fishing community.

Finally, findings in this study may serve as foundation for future researchers, policy makers and health professionals by ensuring that the perceived threats to PrEP acceptability are addressed during the program design and implementation stages.

## Supporting information

**S1 File. Dataset on acceptability of PrEP.**
(XLSX)

**S2 File. REC approval letter.**
(PDF)

**S1 Checklist. Inclusivity in global research.**
(DOCX)

## Acknowledgments

The authors would like to acknowledge the contribution of all the staff of the Department of Public and Community Health, Faculty of Health Sciences, Busitema University, for their support during the design and implementation of the study. We would also like to appreciate the technical support from Dr.Ogwal Daniel, the district health officer Serere district who provided administrative clearance and oversaw data collection processes on the landing site.

## Author contributions

**Conceptualization:** Alex Omoding, Paul Oboth, Joseph K. B. Matovu.

**Data curation:** Ronald Opito, Francis Okello.

**Formal analysis:** Ronald Opito, Francis Okello.

**Investigation:** Alex Omoding, Joseph K. B. Matovu.

**Methodology:** Alex Omoding, Paul Oboth, Joseph K. B. Matovu.

**Project administration:** Alex Omoding, Joseph K. B. Matovu.

**Supervision:** Paul Oboth, Joseph K. B. Matovu.

**Writing – original draft:** Alex Omoding, Ronald Opito.

**Writing – review & editing:** Paul Oboth, Joseph K. B. Matovu.

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
