## [Decision Letter · Decision Letter 0]

18 Mar 2025

PONE-D-24-60531Acceptability of pre-exposure prophylaxis and associated factors among HIV-negative young men in Kagwara fishing community- Serere district: a cross-sectional studyPLOS ONE

Dear Dr. Opito,

Thank you for submitting your manuscript to PLOS ONE. After careful consideration, we feel that it has merit but does not fully meet PLOS ONE’s publication criteria as it currently stands. Therefore, we invite you to submit a revised version of the manuscript that addresses the points raised during the review process.

We look forward to receiving your revised manuscript.

Kind regards,

Adedotun Ogunbajo, Ph.D.

Academic Editor

PLOS ONE

Reviewers' comments:

Reviewer's Responses to Questions

**Comments to the Author**

1. Is the manuscript technically sound, and do the data support the conclusions?

Reviewer #1: Yes

Reviewer #2: Yes

2. Has the statistical analysis been performed appropriately and rigorously? 

Reviewer #1: Yes

Reviewer #2: Yes

3. Have the authors made all data underlying the findings in their manuscript fully available?

Reviewer #1: No

Reviewer #2: Yes

4. Is the manuscript presented in an intelligible fashion and written in standard English?

Reviewer #1: Yes

Reviewer #2: No

5. Review Comments to the Author

Reviewer #1: Congratulations for working on a topic of public importance locally and internationally. You need to work on issue that I have highlighted them in the PDF manuscript file which I have attached.

-The study objectives do not line up well with the title of the study

-The methodology also lacks good clarity and some how is disorganized. Key issues like how calculation of PrEP acceptability was done is not clear.

Reviewer #2: Thank you for the opportunity to review this manuscript. Young men are truly underserved and less is known about their needs concerning HIV prevention.

Below are some comments:

Title: Include Uganda

Key words: Review to HIV; pre-exposure prophylaxis; delete factors, include Uganda

Abstract: Line 22. Which PrEP is being referred to? Oral, Injectable? TDF/FTC, please clarify, also in the main background.

Line 25 include Uganda

Line 28: delete quantitative. Line 31.. Include statistics were

Line 38- include SD in full

Line 45: delete having concern. Wouldn't that be low acceptability/use of PrEP?

Conclusion, what strategies can be recommended for teh challenges identified?

Introduction

Line 54, update to UNSIDS 2024 data and reference

Line 59, include statistics

Line 64, the sentence starting with in addition is not clear whether it refers to communities or people

Line 67, delete sexually active. Those who use PrEP must be those that are sexually active

Line 68, delete high populations who..

Throughout the manuscripts probably refer to high risk young men instead of including HIV

Methods

Line 86- review to this was a cross sectional study.

Re-organize the whole section under sub sections: study design, site, population, data collection, analysis, etc.

Please include a section for participant recruitment to cater for information on how the young men were identified.

Line 95 and following, most of the information should be under participant recruitment. Study setting would include details about Serere and what activities take place in fishing communities, details about health care.

Line 106, provide details of inclusion and exclusion criteria. Exclusion should not necessarily be the opposite of the inclusion criteria.

Line 146, which activities?

Line 150- tested instead of turned?.. those who tested negative were interviewed using....

What is the KOBO tool? Any reference?

Line 158, which variables were fit to be included in the model? How was that determined?

Line 162, what was the level of significance?

For all the results in text, the percentages are actual, using the word about is misleading.

Results

Line 167 SD in full

Line 168 and following, there would be a focus on one side of the proportion instead of listing all.

Table 1: SD should have the Plus/minus symbol

Occupation, clarify on what business, peasant salaries stand for.

Line 176 and following, refer to illicit drug use instead of drugs of abuse/ addiction, etc,

Line 225, bivariate findings are not presented.

Line 237, table footer, what does adjusted for each other mean?

Line 277- review thoughts around window period..

Line 280- review factor was associated with no acceptability...

Line 284- need to provide youth friendly services?

Line 308-309 is a recommendation.

Line 319- among people eligible?

Line 328, any approval from UNCST?

Lines 330-332 are redundant.

Was assent obtained? Did the study recruit emancipated minors? Was parental consent obtained?

Lines 337, not easy to confirm that the place had no one else available. Please review. Line 338-340 is also redundant

Reference: Please include access dates for all web-based references.

Generally review grammar and sentence construction throughout the manuscript.

6. PLOS authors have the option to publish the peer review history of their article (what does this mean? ). If published, this will include your full peer review and any attached files.

**Do you want your identity to be public for this peer review?** For information about this choice, including consent withdrawal, please see our Privacy Policy .

Reviewer #1: No

Reviewer #2: No

---

## [Author Response · Author response to Decision Letter 1]

19 Apr 2025

Response to reviewers and editor have been comprehensively addressed in the document shared here as point-by-point response to reviewers comments

---

## [Editor Report · Decision Letter 1]

16 May 2025

Acceptability of pre-exposure prophylaxis and associated factors among HIV-negative young men in Kagwara fishing community- Serere district, Uganda: a cross-sectional study

PONE-D-24-60531R1

Dear Dr. Opito,

We’re pleased to inform you that your manuscript has been judged scientifically suitable for publication and will be formally accepted for publication once it meets all outstanding technical requirements.

Kind regards,

Adedotun Ogunbajo, Ph.D.

Academic Editor

PLOS ONE
---

## [Editor Report · Acceptance letter]

PONE-D-24-60531R1

PLOS ONE

Dear Dr. Opito,

I'm pleased to inform you that your manuscript has been deemed suitable for publication in PLOS ONE. Congratulations! Your manuscript is now being handed over to our production team.

Kind regards,

on behalf of

Dr. Adedotun Ogunbajo

Academic Editor

PLOS ONE